# Food Ingredients Derived from Lemongrass Byproduct Hydrodistillation: Essential Oil, Hydrolate, and Decoction

**DOI:** 10.3390/molecules27082493

**Published:** 2022-04-12

**Authors:** Luís Rodrigues, Elisabete Coelho, Renata Madeira, Pedro Teixeira, Isabel Henriques, Manuel A. Coimbra

**Affiliations:** 1Laboratório Associado para a Química Verde (LAQV-REQUIMTE), Department of Chemistry, University of Aveiro, 3810-193 Aveiro, Portugal or l.rodrigues@ua.pt (L.R.); or rcm9618@gmail.com (R.M.); mac@ua.pt (M.A.C.); 2Colab4Food—Collaborative Laboratory, 4485-655 Vairão, Portugal; 3União dos Produtores de Refrigerantes de Estarreja Lda (UPREL), 3860-210 Estarreja, Portugal; 4Centre of Environmental and Marine Studies (CESAM), Biology Department, University of Aveiro, 3810-193 Aveiro, Portugal; pedrofteixeira@ua.pt; 5Centre for Functional Ecology, Department of Life Sciences, Faculty of Science and Technology, University of Coimbra, 3000-456 Coimbra, Portugal; ihenriques@ua.pt

**Keywords:** *Cymbopogon citratus*, aromatic plants, antioxidant activity, citral, antimicrobial activity, polysaccharides, matcha tea, hydrosol, terpenes, flavor

## Abstract

Essential oil (EO), hydrolate, and nondistilled aqueous phase (decoction) obtained from the hydrodistillation of lemongrass byproducts were studied in terms of their potential as food ingredients under a circular economy. The EO (0.21%, dry weight basis) was composed mainly of monoterpenoids (61%), the majority being citral (1.09 g/kg). The minimal inhibitory concentrations (MIC) of lemongrass EO against *Escherichia coli*, *Salmonella enterica*, and *Staphylococcus aureus*, were 617, 1550, and 250 μg/mL, respectively. This effect was dependent on the citral content. Particularly for Gram-negative bacteria, a synergism between citral and the remaining EO compounds enhanced the antimicrobial activity. The polymeric material obtained from the nondistilled aqueous phase was composed of phenolic compounds (25% gallic acid equivalents) and carbohydrates (22%), mainly glucose (66 mol%). This polymeric material showed high antioxidant activity due to bound phenolic compounds, allowing its application as a functional dietary fiber ingredient. Matcha green tea formulations were successfully mixed with lemongrass hydrolate containing 0.21% EO (dry weight basis) with 58% of monoterpenoids, being citral at 0.73 g/kg, minimizing matcha astringency with a citrus flavor and extending the product shelf life. This holistic approach to essential oils’ hydrodistillation of *Cymbopogon citratus* byproducts allows for valorizing of the essential oil, hydrolate, and decoction for use as food ingredients.

## 1. Introduction

*Cymbopogon citratus*, Stapf (Lemongrass) is found in countries across Asia, America, Europe, and Africa [1,2]. It is commonly consumed as a herbal infusion or as a premium infusion, where solely lemongrass tips are employed. It contains up to 2% of essential oils on a dry mass basis [3], which is generally recognized as safe (GRAS) by the U.S. Food and Drug Administration (FDA) [4]. Lemongrass’s attractiveness is due to its lemon scent derived from citral, a mixture of the geometric isomers geranial (α-citral), ranging from 50.0 to 27.0%, and neral (β-citral), ranging from 50.8 to 4.5% [1]. The ISO (International Organization for Standardization) provides standards (geranial range 40.0–50.0%, neral range 31.0–40.0%) with which commercial lemongrass EOs should comply [5]. Lemongrass EO from byproducts such as the one used in this study, may not comply with these standards, thus emphasizing the need to identify other applications, such as its use as a flavoring ingredient. The feasibility of lemongrass EO incorporation into beverage formulation has been successfully demonstrated by Kieling et al. [6] who conducted an exhaustive sensorial description of the beverages obtained.

Lemongrass EO has been described as potentially having multiple properties, such as antihypertensive and vasorelaxant [7], antioxidant [8], anti-diabetic and anti-inflammatory [1], antifungal [9], and antibacterial [10]. In fact, citral has been shown to exert a strong antibacterial effect through DNA interaction, on methicillin-resistant *Staphylococcus aureus* in vitro [2,11] and in vivo [12]. Similarly, geranic acid, an acid derived from geranial oxidation and also found in lemongrass EO, is accountable for the inhibition of tyrosinase, possibly preventing phenolic compound oxidation by polyphenoloxidase, responsible for the undesirable browning of foods [13]. Geraniol, a monoterpenol also found in lemongrass EO, has gastric healing properties capable of boosting lemongrass EO’s gastroprotective activity [14]. Thus, despite citral being considered the overall parameter to infer lemongrass EO quality [15,16], other biomolecules may play a role when considering food and beverage applications. These properties support applications of lemongrass EO such as in aromatherapy [17], as an alternative to synthetic pesticides [9], as a mosquito repellent [18], as a natural food preservative [10,19], and as a shelf-life extender [20,21,22].

Hydrodistillation is a widespread method for EO extraction [23], giving rise to an aqueous phase (hydrolate) that distillates together with the essential oil and is physically separated from the essential oil due to its higher polarity and density. This hydrolate is discarded together with the nondistilled aqueous phase and the plant debris. The nondistilled aqueous phase obtained from lemongrass has been reported to be composed mainly of glucose-rich polysaccharides [24] and to have antioxidant and anti-inflammatory properties [25].

In this study, the lemongrass byproduct EO profile will be disclosed as well as its antimicrobial activity against Gram-negative bacteria, namely *Escherichia coli* and *Salmonella anatum* strains, and the Gram-positive bacterium *Staphylococcus aureus*. A holistic strategy will be used to permit the recovery of all fractions (nondistilled aqueous phase and the hydrolate) otherwise wasted and to attribute their possible applications. Under a circular economy concept, the holistic approach to essential oils and the hydrodistillation of *Cymbopogon citratus* byproducts will be completed by exploring lemongrass polysaccharides within the nondistilled aqueous phase regarding their carbohydrate content as well as their antioxidant properties, and by using the hydrolate as a functional ingredient in a matcha tea formulation, complying with beverage safety regulations.

## 2. Results

### 2.1. Characterization of Lemongrass Byproducts: Essential Oil and Hydrolate

The lemongrass EO yield was 0.21%. The same yield was obtained for the lemongrass EO recovered from the hydrolate, resulting in a global yield of 0.42%. These results allow us to hypothesize that these wastes are a relevant source of emulsified EO compounds for possible food applications. To validate this hypothesis, the chemical compositions of the pale-yellow lemongrass EO and the hydrolate EO were determined (Table 1).

Twelve volatile compounds were identified in the lemongrass EO, corresponding to 62.58% of the total oil composition, with citral isomers being the major components (309.21 mg g^−1^ for geranial and 212.17 mg g^−1^ for neral). Other oxygenated compounds were also identified, including geranic acid (36.18 mg g^−1^), geraniol (31.57 mg g^−1^), and neric acid (7.14 mg g^−1^).

Fifteen volatile compounds were identified in the hydrolate EO, corresponding to 59.42% of the total oil composition, with citral isomers being the major components (213.79 mg g^−1^ for geranial and 135.00 mg g^−1^ for neral). Geranic acid (115.20 mg g^−1^), geraniol (69.50 mg g^−1^), and neric acid (28.71 mg g^−1^) were also identified. Although in trace amounts, perillyl alcohol (1.07 mg g^−1^), octanoic acid (3.13 mg g^−1^), and dihydroactinidiolide (1.35 mg g^−1^) were only found in the hydrolate EO.

**Table 1 molecules-27-02493-t001:** Volatile compounds identified in essential oils (EOs) from byproducts of *C. citratus* leaves obtained by hydrodistillation.

Peak Number	Compound	RI_lit_ ^h^	RI_cal_ ^g^	EO ^a^ (mg/g)	Hydrolate EO ^a^ (mg/g)	Water Solubility (mg/L at 37 °C) ^b^	logP	Reliability of ID ^p^
1	Linalool	1507 ^i^	1535	5.03 ± 2.27 *	2.34 ± 1.78 *	480	2.97 ^c^	A, B
2	2-Undecanone	1579 ^j^	1574	3.90 ± 0.50 *	1.54 ± 1.26 *	12	4.25 ^b^	B
3	Neral	1656 ^i^	1650	212.17 ± 55.66 *	135.00 ± 52.79 *	400	3.45 ^d^	B
4	Geranial	1742 ^k^	1704	309.21 ± 58.95 *	213.79 ± 55.57 *	400	3.45 ^d^	B
5	Geranyl acetate	1719 ^i^	1732	5.09 ± 0.83 *	3.60 ± 1.33 *	190	4.48 ^d^	B
6	Citronellol	1762 ^k^	1750	3.57 ± 0.42 *	3.85 ± 0.17 *	350	3.91 ^c^	B
7	Nerol	1836 ^k^	1779	4.48 ± 0.16 *	4.17 ± 0.25 *	1370	3.47 ^e^	B
8	Geraniol	1840 ^k^	1830	31.57 ± 5.95 *	69.50 ± 16.97 *	1370	3.56 ^f^	B
9	Caryophyllene oxide	1999 ^k^	1999	2.54 ± 0.01 *	5.57 ± 1.27 *	7	3.49 ^b^	B
10	Perillyl alcohol	1972 ^i^	2022	-	1.07 ± 0.43	1900	2.50 ^b^	B
11	Octanoic acid	2164 ^l^	2129	-	3.13 ± 1.31	910	2.92 ^b^	B
12	Dihydroactinidiolide	2294 ^m^	2210	-	1.35 ± 0.58	610	3.28 ^b^	B
13	Neric acid	2331 ^k^	2335	7.14 ± 3.51 *	28.71 ± 12.19 *	1220	2.72 ^b^	B
14	Geranic acid	2356 ^n^	2377	36.18 ± 17.34 *	115.20 ± 43.52 *	1220	2.72 ^b^	A, B
15	Palmitic acid	2866 ^o^	2814	4.89 ± 0.73 *	5.38 ± 1.88 *	0.41	6.26 ^b^	A, B
	Total			62.58 ± 10.38	59.42 ± 4.23			
	Yield (% *w*/*w*)			0.21	0.21			

Each value in the table is represented as mean ± standard deviation (*n* = 3); the symbol (*) in the same line indicates a nonsignificant difference (*p* < 0.05). ^a^ Estimated concentrations for all compounds were made by peak area comparisons to the area of a known amount of internal standard (2-undecanol). ^b^ Data obtained from hmdb.ca [26]. ^c^ Data obtained from [27]. ^d^ Data obtained from [28]. ^e^ Data obtained from [29]. ^f^ Data obtained from [30]. ^g^ Retention indices relative to C_14_–C_27_ *n*-alkanes series. ^h^ Retention indices reported in the literature for DB-FFAP columns or equivalent (^i^ [31]; ^j^ [32]; ^k^ [33]; ^l^ [34]; ^m^ [35]; ^n^ [36]; ^o^ [37]). ^p^ The reliability of the identification or structural proposal is indicated by the following: A, mass spectrum and retention time consistent with those of an authentic standard; B, structural proposals are given on the basis of mass spectral data (Wiley 275 Library).

### 2.2. C. citratus Byproducts: EO Antimicrobial Activity

The potential antimicrobial activity of lemongrass EO was investigated by measuring the inhibition zone diameters through an agar disc diffusion assay and assessing the minimal inhibitory concentration (MIC) by a broth microdilution assay (Table 2).

Lemongrass EO showed stronger antimicrobial activity against *S. aureus* (inhibition zone = 13 mm and MIC = 250 μg/mL) than against the Gram-negative bacteria *E. coli* (inhibition zone = 3 mm and MIC = 617 μg/mL) and *S. enterica* (inhibition zone = 0 mm and MIC = 1550 μg/mL). Citral isomers, the major compounds of lemongrass EO, used as standard, followed the same pattern, displaying stronger antimicrobial activity against *S. aureus* (MIC = 105 μg/mL) than against *E. coli* (MIC = 1070 μg/mL) and *S. enterica* (MIC > 2035 μg/mL). Based on these MIC values, due to the lower citral content of the hydrolate EO, much higher MIC values are expected for this sample.

### 2.3. Characterization of Lemongrass Byproducts: NonDistilled Aqueous Phase

Together with the EO, the hydrodistillation process of lemongrass byproducts produced a nondistilled aqueous phase that was recovered, accounting for 14.82 (% *w*/*w*). Carbohydrates accounted for 41%, comprised mainly of glucose (63 mol%) and uronic acids (27 mol%). The free sugars represented 24% of total carbohydrates and were mainly glucose (52%) and fructose (16%) (Table 3) in accordance with the free sugar composition of other aromatic plants [38]. This extract was also composed of phenolics, accounting for 20.7 mg GAE/g allowing an antioxidant activity of 1.4 TEAC (Trolox equivalent antioxidant capacity).

The dialysis of the nondistilled aqueous phase allowed us to obtain the high molecular weight material (HMWM), whose yield was 1.4% of the dried lemongrass byproducts. It was composed of 224 mg/g carbohydrates and enriched with total phenolic compounds (246.3 mg GAE/g), displaying higher antioxidant activity (4.4 TEAC).

The HMWM contained 31.7% of compounds insoluble in cold water. This fraction (WIM) was composed of 259 mg/g sugars, mainly glucose (45 mol%) and uronic acids (34 mol%). The cold-water-soluble material was submitted to an ethanol-graded precipitation, allowing us to obtain a fraction that precipitated in 50% ethanol (Et50), composed of 346 mg/g sugars, glucose (47 mol%), uronic acids (32 mol%), and phenolic compounds (127.7 mg GAE/g), displaying antioxidant activity (3.9 TEAC). The fraction that precipitated in 70% ethanol (Et70) was composed of 313 mg/g sugars, mainly glucose (26 mol%), uronic acids (18 mol%), arabinose (18 mol%), galactose (17 mol%), and phenolic compounds (210.4 mg GAE/g), displaying high antioxidant activity (4.6 TEAC). The fraction soluble in 70% ethanol (EtSn) was composed of 289 mg/g sugars, composed mainly of glucose (76 mol%), uronic acids (7 mol%), rhamnose (6 mol%), and enriched with total phenolic compounds (460.1 mg GAE/g), displaying high antioxidant activity (5.6 TEAC).

### 2.4. Hydrolate as an Ingredient for Beverage Development

The hydrolate derived from the hydrodistillation of lemongrass byproducts, if incorporated into beverages, should be able to provide pleasant flavors (Table 1 and Figure 1). To validate this hypothesis this byproduct was incorporated into a matcha tea formulation to improve its taste acceptability by decreasing the astringency perception and increasing its flavored scents. For this, a matcha tea beverage was prepared with the addition of 20 to 50% (*v*/*v*) of hydrolate (Table 4). The readily prepared beverage containing 35% of hydrolate was the most appealing from a consumer perspective. Beverages containing 30, 20, and 40% were also considered pleasant. The beverages with a percentage higher than 40% were negatively rated due to a strong citrus flavor, whereas those only containing matcha tea were considered astringent.

## 3. Discussion

The overall lemongrass EO yield was 0.42%. As these lemongrass leaves are an industrial byproduct, it was predictable to obtain an inferior yield, yet relevant compared to the 0.73% reported for the lemongrass EO extracted from marketable lemongrass leaves using the same method [3]. Other methods such as microwave-assisted hydrodistillation allow us to obtain a higher yield (1.46%) [39].

The hydrolate EO displays a lower amount of citral isomers (35%) when compared to the lemongrass EO (52%), possibly due to the high concentration and high hydrophobicity of citral (logP = 3.45), which results in a high partition in the oil phase despite its relatively high solubility in water (s = 400 mg/L). Compounds such as geraniol (logP = 3.56; s = 1370 mg/L), nerol (logP = 3.47; s = 1370 mg/L), and other terpenic alcohols such as citronellol (logP = 3.91; s = 350 mg/L), despite their high logP and relatively high solubility in water, did not migrate in great extension to the nonpolar phase due to their very low concentration in the solution (Table 1). The high logP of the geraniol ester geranyl acetate (logP = 4.48) and its low solubility in water (s = 190 mg/L) justifies the fact that it was found relatively more partitioned in the lemongrass EO (Table 1). The acidic compounds such as geranic acid and octanoic acid had concentrations higher in the hydrolate than in the EO, as the carboxylic acid structure confers them a lower logP (2.72 and 2.92, respectively) and a higher solubility in water (1220 and 910 mg/L, respectively).

Due to the relatively high logP and low solubility of myrcene (logP = 4.17; s = 78 mg/L; [26]) and rose furan oxide (logP = 3.74, s = 120 mg/L, [26]), their adsorption on the HS-SPME hydrophobic fiber is favored (Figure 1). As a consequence, these compounds were both found by HS-SPME/GC–MS but not by direct GC-MS injection, in accordance with the literature [2,40]. On the contrary, geranic acid is poorly sorbed by the HS-SPME fiber due to its relatively lower logP (Table 1).

The results obtained showed that it is possible to obtain from lemongrass byproducts two aroma-rich products: the EO and the hydrolate. The latter is a relevant source of emulsified EO compounds with a similar volatile profile to lemongrass EO, rendering it a suitable ingredient for food applications.

Both lemongrass EO and citral showed stronger antimicrobial activity against *S*. *aureus* than against the Gram-negative bacteria *E*. *coli* and *S*. *enterica*. These results are in agreement with the literature supporting higher EO resistance of Gram-negative bacteria, attributed to the presence of hydrophilic lipopolysaccharides in the outer membrane [41], rendering them resistant to lipophilic compounds. On the other hand, deprived of an outer membrane, Gram-positive bacteria are more easily affected by EO hydrophobic constituents which increase ion permeability and leakage of vital intracellular content [42]. It can be hypothesized that lemongrass EO has the potential to be used for similar applications to those already available for premium lemongrass EO [43], namely due to its antimicrobial activity. Citral isomers, used as standard, displayed slightly stronger antimicrobial activity against *S*. *aureus* (MIC = 105 μg/mL) than the lemongrass EO (130 μg citral/mL) (Table 2). The close range of MIC values suggests that citral isomers are the active EO compounds against *S*. *aureus*. On the other hand, for Gram-negative bacteria, citral isomers displayed clearly higher MIC values (1070 μg/mL and >2035 μg/mL for *E*. *coli* and *S*. *enterica*, respectively) than the lemongrass EO (322 μg citral/mL and >808 μg citral/mL, respectively). The lower MIC values obtained for the lemongrass EO suggest a synergism between citral and other lemongrass EO components. This effect may be due to the presence of geraniol. In fact, citral isomers and geraniol have synergistic activity against *Xanthomonas citri* subsp. *citri*, a Gram-negative bacterium, explained by the interaction of these compounds with the outer membranes of these bacteria, allowing their permeation into the cell [44].

Together with EOs, the hydrodistillation process of lemongrass byproducts produced a nondistilled aqueous phase (decoction) which was recovered accounting for 14.8% (*w*/*w*). This yield was higher than the 10.9% previously reported for premium lemongrass, using a longer extraction time (3 h) and the same sample-to-water ratio (1:15 *w*/*v*) [25]. The HMWM, whose yield was 1.4%, was composed of 224 mg/g carbohydrates and enriched with total phenolic compounds (246.3 mg GAE/g), displaying higher antioxidant activity (4.4 TEAC). These results suggest that phenolic compounds were bound to polymeric carbohydrates rich in glucose (66%), arabinose (9%), and uronic acids (8%). The high abundance of glucose in the WIM and Et50 fractions may indicate the presence of starch, which can occur in leaves [45], due to its insolubility in cold water and in ethanol solutions. The high abundance of glucose in the EtSn (76%) may derive from polymeric phenols, as observed in the HMWM that migrates from cork stoppers [46]. The presence of a high percentage of uronic acids in Et50 and Et70, together with arabinose, galactose, and rhamnose, mainly in Et70, allowed us to infer the presence of pectic polysaccharides, highly branched in Et70 [47]. The Et70 fraction also contains high antioxidant activity, although only presenting half of the phenolic compound content of the EtSn fraction, possibly due to the phenolic acids [48]. It suggests that phenolic compounds with greater antioxidant activity are present in Et70 rather than in the EtSn. These results are in accordance with the occurrence of phenolic compounds covalently linked with highly branched pectic polysaccharides [47]. This study showed that the nondistilled aqueous phase from lemongrass hydrodistillation is a source of food ingredients as functional polysaccharides with relevant antioxidant activity.

Pure matcha tea presents molds and yeasts (56 CFU/mL) immediately after its preparation as a result of a warm decoction. Nevertheless, the value of 56 CFU/mL is below the maximum admissible for molds and yeasts at 25 °C (10^3^ CFU/mL) [49]. The incorporation of hydrolate prevents the proliferation of fungi after 79 days of beverages storage (Table 4), showing that the hydrolate also displays antifungal activity. Citral isomers as well as geraniol have been reported to inhibit eukaryotes’ ATP-dependent molecular chaperone (HSP90-ATPase) involved in protein folding [50]. This may explain the antifungal activity of lemongrass EO against *Fusarium oxysporum* f.sp. *lycopersici* spores [9] and the elimination of molds and yeasts when incorporating hydrolate into matcha tea beverages.

Considering the growth of total mesophiles, a slight proliferation in this period was observed. Nevertheless, these values are far below the maximum values for total mesophiles at 25 °C (10^5^ CFU/mL) [49]. These results demonstrate the active role of the lemongrass essential oil contained in the hydrolate, allowing this byproduct to be used to create novel healthy beverages with a pleasant aroma and an increased shelf life. According to FEMA (Flavor and Extract Manufacturers Association of the United States), the average and maximum use levels of citral for nonalcoholic beverages is 17 ppm and 28 ppm, respectively [51]. The matcha tea beverage containing 20% hydrolate displayed a citral level (31 ppm) slightly higher than the maximum citral level recommended by FEMA, while being evaluated as pleasant. Thus, the results obtained encourage the application of lemongrass hydrolate in selected beverage categories to modulate astringency, namely in no-sugar-added beverages, and to extend shelf-life.

## 4. Materials and Methods

### 4.1. Collection of Plant Material of Cymbopogon citratus

Dried leaves agro-industrial byproduct of *Cymbopogon citratus* (DC) Stapf were provided by Ervital-Infusões e Condimentos Biológicos and were farmed according to organic standards in greenhouse facilities (40°58′30″ N 7°54′09″ W, altitude 910 m) in Montemuro Montain, Mezio, Castro Daire, Portugal. This agro-industrial byproduct was the first crop collected. It was considered a byproduct because it presented nonmarketable brown spots. The plant material was transported in dark bags and stored in the dark at room temperature until further analysis. All reagents used in the analysis were of analytical grade. Citral standard was purchased from Sigma-Aldrich (St. Louis, MO, USA).

### 4.2. Extraction of Essential Oils (EOs) and Polymeric Material

Dried leaves of lemongrass (100 g) were subjected to a simple steam distillation with distilled water (1:15 *w*/*v*) in a round-bottom flask. The flask was then kept in a heating mantle boiling at 100 °C for 2.5 h. Lemongrass EO was physically separated in a U tube and dried using anhydrous sodium sulphate. The hydrolate was collected in a beaker through the U tube siphon. To obtain the EO present in the hydrolate, it was extracted three times with 1 volume of dichloromethane to 5 volumes of hydrolate and dried over anhydrous sodium sulphate. Dichloromethane was removed by nitrogen stream. The oils were stored in sealed vials at 4 °C until GC–MS analysis. Hydrodistillation was performed three times obtaining three independent hydrolates and EOs. The hydrodistillation also resulted in a nondistilled aqueous phase which was filtered, rota-evaporated, freeze-dried, and dialyzed (cut-off 12–14 kDa) to obtain the HMWM, which is the dialysis retentate, containing compounds with a molecular weight superior to 12–14 kDa.

To further purify lemongrass HMWM, a graded precipitation with ethanol was performed. Lemongrass HMWM was dissolved in water (200 mg in 20 mL, final concentration 10 mg/mL) and precipitated by the addition of absolute ethanol [52]. Ethanol was added to obtain a 50% (*v*/*v*) aqueous solution. The aqueous solution was stirred for 2 h at 4 °C, centrifuged, and the precipitate at 50% ethanol concentration (Et50) was obtained. Ethanol was added to the supernatant to obtain a 70% (*v*/*v*) aqueous solution. This aqueous solution was stirred for 2 h at 4 °C, centrifuged, and the precipitate at 70% ethanol concentration (Et70) was removed from the supernatant solution (EtSn). In order to remove the ethanol completely, each precipitate was water-dissolved and rota-evaporated, frozen, and freeze-dried until further analysis.

### 4.3. Matcha Tea Preparation

To obtain the matcha tea concentrate, 12.5 g of Powder Matcha Bio Shine powder and 0.625 g of anhydrous citric acid (E330) were added to 1 L of preheated water, keeping the suspension under stirring (700 rpm) with a magnetic bar at 75–80 °C for 5 min, using a water bath. The concentrate was subsequently cooled with 1.5 L of water at room temperature. Matcha tea beverages containing hydrolate were readily prepared by stirring together the prepared matcha tea concentrate with the hydrolate in proportions as described in Table 4. The pH of the resulting solutions was then measured.

Sensory analysis was performed from a consumer preference perspective, using 5 people (3 females, 2 males, with ages between 23 and 55 years old, all working for the beverage company). After tasting the beverages, they scaled the product as (−) unpleasant, (+) acceptable, (++) pleasant, and (+++) very pleasant.

### 4.4. Determination of Essential Oils Composition

Lemongrass byproduct EOs and the hydrolate obtained were analyzed on an Agilent Technologies 6890 N Network gas chromatograph (from Agilent Technologies, Inc., Santa Clara, CA, USA), equipped with a 30 m × 0.32 mm (I.D.), 0.25 μm of film thickness DB-FFAP fused silica capillary column (J&W Scientific Inc., Folsom, CA, USA), connected to an Agilent 5973 quadrupole mass selective detector (MS, Agilent Technologies, Inc., Santa Clara, CA, USA). Identification of volatile compounds was achieved by comparison of the GC retention times and mass spectra with those, when available, of the pure standard compounds. All mass spectra were also compared with the library data system of the GC-MS equipment (Wiley 275) and according to the compounds previously described for the plant EOs. The identification was also supported by the experimentally determined retention index (RI) values that were compared, when available, with the values reported in the literature for chromatographic columns similar to the one used in this work. For quantification purposes, 2-undecanol (49 μL) was added as internal standard along with dichloromethane (150 μL) and EO (1 μL) prior to analysis. All measurements were made with three replicates, each replicate representing the analysis of one different aliquot (1 μL) of each EO sample, which was injected directly into the GC [53].

The volatile profile of hydrolate samples was also assessed by headspace solid-phase microextraction HS-SPME/GC–qMS. For each HS-SPME assay, 3 mL of hydrolate sample was placed into 10 mL glass vial, with 0.6 g of sodium chloride, and the vial was capped. The SPME fiber (1 cm stable-flex^TM^ fused silica fiber, coated with partially cross-linked 50/30 μm DVB/CAR/PDMS) was manually inserted into the sample headspace vial for 10 min at 40 °C with constant stirring. Then, the SPME fiber was manually inserted into the GC injection port at 250 °C and kept for 3 min for desorption. The injection port was lined with a 0.75 mm (I.D.) splitless glass liner. Splitless injections were used. The oven temperature program was as follows: initial temperature was 40 °C with a linear increase of 5 ⁰C/min up to 220 °C, followed by linear increase of 10 °C/min until 250 °C, remaining thus until the end of the run (42 min) [54].

### 4.5. Sugar Analysis

The sugar composition of the polysaccharides was assessed by gas chromatography-flame ionization detection (GC-FID) as alditol acetates and quantified using 2-deoxyglucose (200 μL) as internal standard following the general procedure described by Bastos et al. [55]. The hydrolysis was performed in 1 M H_2_SO_4_ at 100 °C for 2.5 h. After 1 h hydrolysis, 0.5 mL were collected to quantify the uronic acids. Monosaccharides were reduced with NaBH4 and acetylated, with acetic anhydride and 1-methylimidazole as catalyst. The alditol acetates were analyzed by a GC-FID PerkinElmer-Clarus 400 with a capillary column DB-225 (30 m length, 0.25 mm inner diameter, and 0.15 µm film thickness). The oven temperature program was as follows: 200 to 220 °C at a rate of 40 °C/min (7 min), increasing to 230 °C at a rate of 20 °C/min (1 min). The temperatures of injector and detector were 220 and 230 °C, respectively. Hydrogen was used as carrier gas at a flow rate of 1.7 mL/min. Free sugars were determined without the hydrolysis step. Fructose was quantified from the ratio that epimerized to mannitol (43%) during the reduction step [56]. Free glucose was determined by the difference between the total glucitol detected and the glucitol yielded by fructose reduction. Uronic acid content was determined by the *m*-phenylphenol colorimetric method using galacturonic acid as standard [55].

### 4.6. Antimicrobial Activity of EOs

The antimicrobial activity of EOs was evaluated using two Gram-negative bacterial strains (*Escherichia coli* ATCC 25922 and *Salmonella enterica* sv Anatum SF2) and one Gram-positive bacterial strain (*Staphylococcus aureus* ATCC 6538). *S*. *enterica* sv Anatum SF2 was isolated from seagull feces on the island of Berlengas (Peniche, Portugal) [57], whereas *E*. *coli* ATCC 25922 and *S*. *aureus* ATCC 6538 were obtained from the American Type Culture Collection [53].

For disc diffusion assays, cell suspensions were prepared according to Santos et al. [53] and used to inoculate Mueller Hinton agar (Merck, Darmstadt, Germany) plates with a sterile swab. Sterile 6 mm filter paper discs (Liofilchem, Italy) were placed on the plates and impregnated with sterile water and EOs. Commercial discs containing antibiotics (Oxoid, UK) were used as positive controls, namely ciprofloxacin (5 μg) and gentamicin (10 μg). Negative controls consisted of paper discs with sterile water or dimethyl sulfoxide (DMSO).

After incubation at 37 °C for 18–24 h, the diameter of the inhibition zone was measured in millimeters. The assay was carried out three times for each strain. The amount of lemongrass EO added to each test disc was the same (1 µL) for all assays for each microorganism.

The minimal inhibitory concentration (MIC) values were determined using a broth microdilution assay [53]. Aliquots of lemongrass EO were dissolved in 1% DMSO and tested in maximum stepwise increments of 25 µg/mL in Mueller–Hinton broth (MH broth; Merck). 96-Well plates were prepared by dispensing in each well 5 μL of the microorganism suspensions (optical density of microorganism suspensions at 625 nm between 0.09 and 0.13), aliquots of lemongrass EO or citral standard containing 2 μL DMSO emulsified in MH broth, and MH broth to reach 200 μL per well final volume. Negative controls were used with 5 μL of the microorganism suspensions and 2 μL DMSO in 193 μL MH broth. Microbial growth in each medium was determined by measuring the optical density at 600 nm, after incubation at 37 °C for 18–24 h. EOs were tested three times against each organism.

### 4.7. Total Phenolic Compounds

Total phenols were assessed by the Folin–Ciocalteu colorimetric assay as described in Singleton et al. [58]. Aliquots of samples (15 µL) were used in the concentration range of 0.1 to 1.2 mg/mL and were assessed by their reactivity with Folin–Ciocalteu reagent (15 µL) and sodium carbonate at 70 mg/mL (150 µL). The reaction mixtures (240 µL) were incubated at 30 °C for 60 min. The absorbance values were measured at 750 nm. A calibration curve of gallic acid, at concentrations ranging from 0.030 to 0.30 mg/mL, was built and the results were expressed as gallic acid equivalents (mg GAE per g of dry matter) using the equation y = 288.92x + 0.0439 (r^2^ = 0.995) for nondistilled aqueous phase and HMWM samples, whereas equation y = 330.78x + 0.0594 (r^2^ = 0.998) was used for ethanolic precipitation-derived fractions. The data were expressed as mean ± standard deviation (SD) of nine replicas.

### 4.8. ABTS^•+^ Radical Cation Decolorization Assay

In vitro, antioxidant capacity was evaluated according to 2,2′-azinobis-(3-ethylbenzothiazoline-6-sulfonic acid) (ABTS^•+^) method [59]. For this, 6.6 mg of potassium persulfate were mixed with 5 mL of ABTS^+^ solution (3.8 g/L) and placed in the dark for 12 h. The obtained ABTS^•+^ solution was diluted in distilled water until an absorbance value of approximately 0.70 was reached at 734 nm. The prepared extracts were diluted in water, and 50 µL of the solution were mixed with 250 µL of ABTS^•+^ solution and stored in the dark for 20 min. Afterward, the absorbance was measured at 734 nm. A standard calibration curve was prepared with Trolox (6-Hydroxy-2,5,7,8-tetramethylchroman-2-carboxylic acid), at concentrations ranging from 0 to 105 μg/mL, and the TEAC was measured as mM Trolox eq./g dry matter using the equation y = −14.78x + 0.5673 (r^2^ = 0.996) for aqueous extract and HMWM samples and the equation y = −18.104x + 0.7361 (r^2^ = 0.993) for ethanolic precipitation-derived fractions. The results were expressed in mM Trolox equivalents/g of sample.

### 4.9. Microbial Analyses of Matcha Tea Beverages

Microbiological analysis of matcha tea beverages was performed using the method adopted by the beverage company Uprel, Lda, Portugal. To determine total mesophiles, samples were inoculated on CompactDry^TM^ TC chromogenic plates, containing nutrient agar medium supplemented with the redox indicator tri-phenyltetrazolium chloride, in which total mesophiles form red colonies, thus allowing for their counting. The incubation at 30 °C occurred for 48 h according to ISO 4833-1: 2013 standard.

To determine molds and yeasts, samples were inoculated on CompactDry^TM^ YM plates (Ambifood) containing an X-Phos chromogenic enzyme substrate, in which molds and yeasts can be differentiated by color development. Most yeasts develop a blue color and molds appear as cottony colonies with characteristic colors. The procedure is similar to that used to determine total mesophiles. The incubation had a period of 3 to 7 days at 25 °C and the procedure was followed according to ISO 7959:1987 standard.

### 4.10. Statistical Analyses

Statistical analyses were performed in order to compare the antimicrobial effects of *C*. *citratus* byproduct EOs, being considered statistically significant when *p* < 0.05. One-way analysis of variance (ANOVA) was performed followed by a multiple comparison test (Tukey’s HSD) using the GraphPad Prism version 5.00 for Windows (trial version, GraphPad Software, San Diego, CA, USA). It was also performed to compare the volatile profiles of *C*. *citratus* EOs, being considered statistically significant when *p* < 0.05. Student’s *t*-test was performed followed by Welch-s correction using the same software.

## 5. Conclusions

*C. citratus* leaf byproducts are a source of citral (1.83 g/kg), geranic acid (0.32 g/kg), and geraniol (0.21 g/kg), among other terpenic compounds. This lemongrass EO showed antibacterial activity against Gram-positive and Gram-negative bacteria, presenting MIC values of 250 μg/mL against *S. aureus*, 617 μg/mL against *E. coli*, and 1550 μg/mL against *S. enterica*. Regarding the Gram-negative bacteria studied, a synergism between citral and the remaining EO compounds enhanced the antimicrobial activity. Upon hydrodistillation to recover the EO, it was possible to obtain a nondistilled aqueous phase (decoction) composed of free sugars, phenolic compounds, and polysaccharides with antioxidant activity, with possible application as a functional dietary fiber in the food industry. The resultant hydrolate is composed of emulsified citral-rich EO with the potential to be successfully used in matcha tea formulations by providing taste and extended shelf life. Overall, lemongrass leaf byproducts can be potentially applied in sectors where lemongrass premium leaves find application, thus contributing to a circular economy, a driving force for sustainability.

## Figures and Tables

**Figure 1 molecules-27-02493-f001:**
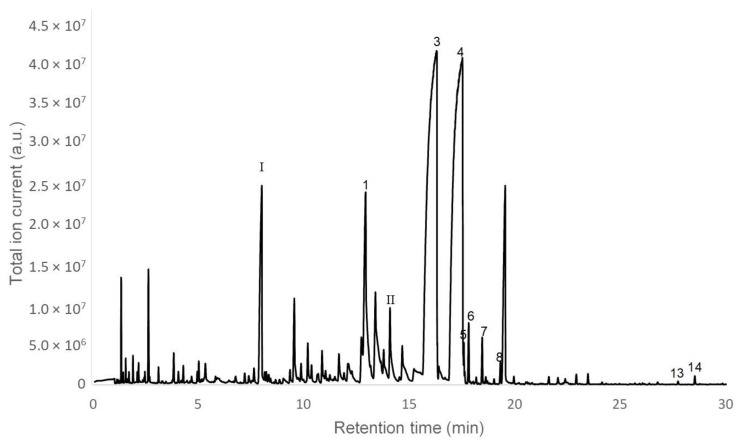
HS-SPME/GC–MS chromatogram of the volatile composition of lemongrass byproduct hydrolate. Numbers above correspond to the peak numbers from Table 1. I—Myrcene. II—Rose furan oxide. The reliability of the identification is given on the basis of mass spectral data (Wiley 275 Library).

**Table 2 molecules-27-02493-t002:** Inhibition zones (mm) and minimal inhibition concentration (μg/mL) of *C. citratus* byproduct EO.

Bacterial Strains	Inhibition Zone				MIC		
	*C. citratus* EO ^a^	CIP ^b^	Gen ^c^	Sterile Water	*C. citratus* Byproducts EO (μg/mL)	Citral Standard (μg/mL)	*C. citratus* Byproducts EO (μg citral/mL)
*E. coli* ATCC 25922	3 ± 5 * (0.91 mg/0.33 mg)	33 ± 1 ^#^	22 ± 0 ^$^	ND	617 ± 31.2 *	1070 ^#^	322 ± 19.9 *
*S. enterica* sv Anatum SF2	0 * (0.91 mg/0.33 mg)	31 ± 1 ^#^	18 ± 2 ^$^	ND	1550 ± 20.4 *	>2035 ^#^	808 ± 13.0 ^$^
*S. aureus* ATCC 6538	13 ± 2 * (0.91 mg/0.33 mg)	26 ± 2 ^#^	20 ± 0 ^$^	ND	250 *	105 ^#^	130 ^$^

Each value in the table is represented as mean ± standard deviation (*n* = 3); different symbols (*, ^#^, ^$^) in the same line indicate significant difference (*p* < 0.05). ^a^ inhibition zone in diameter around the discs impregnated with the amount of essential oil/amount of citral in the essential oil described in parenthesis for each microorganism. ^b^ CIP, ciprofloxacin (5 μg). ^c^ GEN, gentamicin (10 μg). ND, no inhibitory effect was detected.

**Table 3 molecules-27-02493-t003:** Carbohydrate composition, total phenolic content, and Trolox equivalent antioxidant capacity from lemongrass nondistilled aqueous phase (free sugars and total sugars), HMWM lemongrass, and fractions (WIM, Et50, Et70, and EtSn) obtained after ethanol precipitation of HMWM.

Samples	Yield (%)	Total Carbohydrates (mg/g)	Carbohydrates (%Molar)								Total Phenolics ^a^ (mg GAE/g)	TEAC ^a^ (mM Trolox eq./g)
			Rha	Ara	Xyl	Man	Fru	Gal	Glc	UA		
Free sugar	14.8 *	97	9	6	0	0	16	9	52	-	20.7 ± 2.9	1.4
Total sugar		407	2	1	tr	3	-	3	63	27		
HMWM	1.4 *	224	5	9	4	1	-	7	66	8	246.3 ± 18.4	4.4
WIM	31.7	259	2	8	3	1	-	7	45	34	-	-
Et50	9.2	346	3	6	2	1	-	9	47	32	127.7 ± 3.0	3.9
Et70	7.9	313	5	18	14	2	-	17	26	18	210.4 ± 10.0	4.6
EtSn	38.3	289	6	5	3	1	-	3	76	7	460.1 ± 49.4	5.6

Carbohydrates (%molar) are presented as average of 2 replicates from sugar analysis for all the above lemongrass samples. Rha: rhamnose, Ara: arabinose, Xyl: xylose, Man: mannose, Gal: galactose, Glc: glucose, UA: uronic acids. * Yields refer to the 100 g lemongrass leaves hydrodistilled. The remaining yields abovementioned were calculated relative to the pristine weight before ethanolic fractionation. WIM (water-insoluble material). Tr: traces. ^a^ Values are mean ± standard deviation of three replicates.

**Table 4 molecules-27-02493-t004:** Consumer sensory evaluation and microbial analysis of matcha tea beverages, after 79 days brew preparation, containing different percentages of hydrolate from lemongrass byproduct hydrodistillation.

% Hydrolate	Consumer Evaluation	Citral (ppm)	pH	Total Mesophiles (CFU/mL)	Moulds and Yeasts (CFU/mL)
0	−	0	5.99	0	>10^3^
20	+	31	5.82	26	0
30	++	47	nd	nd	nd
35	+++	54	5.92	20	0
40	+	62	nd	nd	nd
45	−	70	nd	nd	nd
50	−	78	6.20	0	0

Consumer evaluation (5 People) was rated in (−) unpleasant, (+) acceptable, (++) pleasant, and (+++) very pleasant. CFU: Colony Forming Units; nd: not determined.

## Data Availability

Not applicable.

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
