# Peer review of "Food Ingredients Derived from Lemongrass Byproduct Hydrodistillation: Essential Oil, Hydrolate, and Decoction"

_molecules, 2022, doi:10.3390/molecules27082493_

Round 1

Reviewer 1 Report

The manuscript entitled "Food ingredients derived from lemongrass by products hydro-distillation: essential oil, hydrosol, and decoction" represent very interesting research, and brings a multitude of research data that will be very useful in future studies.

As part of the Material and Methodology section,

4.1. Collection of Plant Material of Cymbopogon citratus

Authors must add more information. Where was the crop grown? Conditions? Who determines species? Where was it deposited? 

4.2. Extraction of essential oils (EOs) and polymeric material

...yielding two distinct essential oils... this is not correct! During essential oil distillation, it obtains essential oil and its byproduct - hydrolate. Further, hydrolate could be used for secondary extraction in order to obtain a recovery of essential oil. Please correct and reformulate this part.

In addition, please specify which apparatus you use for essential oil extraction! How did you collect the hydrolate? How many hydrolate did you obtain during this process?

4.4 and 4.5 subtitles are the same! Subtitles need to be corrected.

Author Response

Answers to the Reviewer comments

Manuscript ID:  molecules-1666183

Food ingredients derived from lemongrass byproducts hydro-distillation: essential oil, hydrolate, and decoction

Luís Rodrigues, Elisabete Coelho, Renata Madeira, Pedro Teixeira, Isabel Henriques and Manuel A. Coimbra

-Reviewer #1

Comment 1:

“As part of the Material and Methodology section,

4.1. Collection of Plant Material of Cymbopogon citratus

Authors must add more information. Where was the crop grown? Conditions? Who determines species? Where was it deposited?

Authors answer 1:

The authors thank the reviewer for the suggestions which help to clarify the manuscript. The required information was added in section 4.1 as follows:

Dried leaves agro-industrial byproduct of Cymbopogon citratus (DC) Stapf were provided by Ervital - Infusões e Condimentos Biológicos and were farmed according to organic standards in greenhouse facilities (40°58’30’’N 7°54’09’’W, altitude 910 m) in Montemuro Montain, Mezio, Castro Daire, Portugal. This agro-industrial byproduct was the first crop collected. It was considered a byproduct because it presented non-marketable brown spots. As a byproduct from an aromatic herbal industry, it is not used to the deposition in an herbarium. The specimen has been provided to the University of Aveiro herbarium.

Comment 2:

“4.2. Extraction of essential oils (EOs) and polymeric material

...yielding two distinct essential oils... this is not correct! During essential oil distillation, it obtains essential oil and its byproduct - hydrolate. Further, hydrolate could be used for secondary extraction in order to obtain a recovery of essential oil. Please correct and reformulate this part.

In addition, please specify which apparatus you use for essential oil extraction! How did you collect the hydrolate? How many hydrolate did you obtain during this process?”

Authors answer 2:

Authors replaced “hydrosol” for “hydrolate” on the entire manuscript to follow the suggestion.

Simple steam distillations were used.

Lemongrass EO was physically separated from the hydrolate using a U tube and a siphon to allow the collection of the hydrolate in a beaker.

Each hydrodistillation allowed to obtain one hydrolate. Since hydrodistillation was performed three times, it was obtained three independent hydrolates.

This information was added to Material and Methods section.

Comment 3:

“4.4 and 4.5 subtitles are the same! Subtitles need to be corrected.”

Authors answer 3:

The Section 4.5 was corrected to “Sugar analysis”.

Reviewer 2 Report

Abstract:

- The chemical composition of Hydrosol EO was also evaluated and must be presented in the abstract;

- - Why was the minimal inhibitory concentration (MIC) of Hydrosol EO not evaluated against Escherichia coli, Salmonella enterica, and Staphylococcus aureus?

- Why did the authors choose to use only Hydrosol EO in the formulation of Matcha green tea and not EO? Justify.

Results:

No need to display the chromatogram

Pag. 2, Line 191: "The readily prepared beverage containing 35% of hydrosol was the most appealing from a consumer perspective." Was sensory analysis performed? You have to add the methodology.

Material and methods:

Page 5 Line 310:  It was not clear whether the single 100g of the leaf sample was extracted three times for 2.5h each, totaling 7.5h of extraction, or were three 100g samples used for three extractions?

Page 5 Line 312: How was the physical separation of  EO made? Describe.

Page 5, L338: "Matcha tea beverages containing hydrosol were readily prepared by stirring together both components."

- It was not clear which second component Matcha tea was added and how much.

- Check the title of item 4.5. Determination of essential oils composition. Should be "Determination of the polysaccharides of non-distilled aqueous phase"?

- Justify why the antimicrobial activity was evaluated only for OEs and not for all extracts.

Author Response

Answers to the Reviewer comments

Manuscript ID:  molecules-1666183

 Food ingredients derived from lemongrass byproducts hydro-distillation: essential oil, hydrolate, and decoction

Luís Rodrigues, Elisabete Coelho, Renata Madeira, Pedro Teixeira, Isabel Henriques and Manuel A. Coimbra

Reviewer #2

Comment 1:

“Abstract: the chemical composition of Hydrosol EO was also evaluated and must be presented in the abstract”

Authors answer 1:

   The authors thank the reviewer for the suggestions which help to improve the manuscript. The chemical composition of Hydrolate EO was included in the abstract.

Comment 2:

“Why was the minimal inhibitory concentration (MIC) of Hydrosol EO not evaluated against Escherichia coli, Salmonella enterica, and Staphylococcus aureus?”

Authors answer 2:

    The low citral content of hydrolate EO in comparison with that obtained by physical separation suggests a higher MIC value against the tested microorganisms. For that reason, only physically separated lemongrass byproducts EO obtained by physical separation was tested. A comment was included in the Results section.

Comment 3:

“Why did the authors choose to use only Hydrosol EO in the formulation of Matcha green tea and not EO? Justify.”

Authors answer 3:

    The incorporation of lemongrass EO would require dilution and could give rise to phase separation challenges. Choosing to use only the hydrolate for the Matcha green tea formulations, it allowed a more sustainable approach without adding water. Plus, the emulsified nature of the hydrolate did not put any phase separation challenges.

Comment 4:

“Results: no need to display the chromatogram”

Authors answer 4:

    The authors consider useful to display the chromatogram since it provides insight about the volatile composition in the headspace of the beverage. In a consumer´s perspective, it gives the volatile compounds that can be perceived, which complements the data of Table 1.

Comment 5:

“Pag. 2, Line 191: "The readily prepared beverage containing 35% of hydrosol was the most appealing from a consumer perspective." Was sensory analysis performed? You have to add the methodology.”

Authors answer 5:

    Sensory analysis was performed in a consumer preference perspective using 5 people (3 females, 2 males, with ages between 23-55 years old, all belonging to the beverage company). After tasting the beverages, they scaled the product as: (-) unpleasant, (+) acceptable, (++) pleasant, and (+++) very pleasant. This information was added to the section 4.3.

Comment 6:

“Material and methods: Page 5 Line 310:  It was not clear whether the single 100g of the leaf sample was extracted three times for 2.5h each, totaling 7.5h of extraction, or were three 100g samples used for three extractions?”

Authors answer 6:

Each hydrodistillation allowed to obtain one hydrolate. Since hydrodistillation was performed three times, it was obtained three independent hydrolates.

This information was added to Material and Methods section.

Comment 7:

“Page 5 Line 312: How was the physical separation of EO made? Describe.”

Authors answer 7:

    Lemongrass EO was physically separated by means of a U tube and removed using a pipette. This information was added to Material and Methods section.

Comment 8:

“Page 5, L338: "Matcha tea beverages containing hydrosol were readily prepared by stirring together both components.

- It was not clear which second component Matcha tea was added and how much."

Authors answer 8:

The sentence was improved in order to clarify the Matcha tea preparation:

Matcha tea beverages containing hydrolate were readily prepared by stirring together both components, the prepared matcha tea concentrate with the hydrolate in proportions as described in Table 4.

Comment 9:

“Check the title of item 4.5. Determination of essential oils composition. Should be "Determination of the polysaccharides of non-distilled aqueous phase"?”

Authors answer 9:

Section 4.5 title was corrected as “Sugar analysis”

Comment 10:

“Justify why the antimicrobial activity was evaluated only for OEs and not for all extracts.”

Authors answer 10:

The terpenic compounds are currently associated to antimicrobial activity. For this reason, EO rich in terpenic compounds were evaluated for this activity. The extracts from non-distilled aqueous phase were rich in polysaccharides, some bounded to phenolic compounds, which was considered to have more potential as antioxidant dietary fibers. Moreover, polysaccharides are not associated to have antimicrobial activity.

Reviewer 3 Report

This manuscript may be deserved for its publication in Molecules but some points should be explained by the Authors.

  1. The same numerical values presented in Table 1 should be included in the presented results in the section 2.1.
  2. What exactly does the information provided by the Authors under Table 1 - "different symbols (*) in the same line indicate a significant difference" mean? Please explain.
  3. The chromatogram presented by the Authors in Figure 1 probably corresponds to Table 1 and not to Table 2.
  4. The Authors obtained essential oil, hydrosol and decoction by hydrodistillation of lemongrass dried leaves. The method of obtaining these isolates should be described more clearly in section 4.2. or there should be information in brackets about which part of the description relates to the received isolates.
  5. In which apparatus (Derynga or Clevengera) was the hydrodistillation of dried lemongrass leaves carried out? (line 310)
  6. In what weight ratio were the components mixed to prepare matcha tea beverages containing hydrosol? (line 339)
  7. Both section 4.4. and section 4.5. has been entitled by the Authors as "Determination of essential oils. Composition of which essential oils have been specified by the Authors in section 4.5?
  8. Please provide information about the solvent in which the essential oils were dissolved before they were applied to the paper discs in order to determine the zone of inhibition of the growth of the tested bacterial strains? (line 406)

Author Response

Answers to the Reviewer comments

Manuscript ID:  molecules-1666183

Food ingredients derived from lemongrass byproducts hydro-distillation: essential oil, hydrolate, and decoction

Luís Rodrigues, Elisabete Coelho, Renata Madeira, Pedro Teixeira, Isabel Henriques and Manuel A. Coimbra

Reviewer #3

Comment 1:

“The same numerical values presented in Table 1 should be included in the presented results in the section 2.1.”

Authors answer 1:

  The authors thank the reviewer for the suggestions which help to clarify the manuscript.  The authors followed the suggestion.

Comment 2:

“What exactly does the information provided by the Authors under Table 1 - "different symbols (*) in the same line indicate a significant difference" mean? Please explain.”

Authors answer 2:

Equal symbols in the same line indicate lack of significant difference between EO and the hydrolate EO. This information was improved in the footnote text of Table 1.

Comment 3:

“The chromatogram presented by the Authors in Figure 1 probably corresponds to Table 1 and not to Table 2.”

Authors answer 3:

    The correction was performed accordingly.

Comment 4:

“The Authors obtained essential oil, hydrosol and decoction by hydrodistillation of lemongrass dried leaves. The method of obtaining these isolates should be described more clearly in section 4.2. or there should be information in brackets about which part of the description relates to the received isolates.”

Authors answer 4:

The information provided was improved.

Comment 5:

“In which apparatus (Derynga or Clevengera) was the hydrodistillation of dried lemongrass leaves carried out? (line 310)”

Authors answer 5:

Simple steam distillations were used. The experimental procedure was improved, including this information.

Comment 6:

“In what weight ratio were the components mixed to prepare matcha tea beverages containing hydrosol?”

Authors answer 6:

 The components to prepare matcha tea beverages containing hydrolate were mixed in proportions indicated in the column 1 of table 4. The description of the preparation of the beverage was improved.

Comment 7:

“Both section 4.4. and section 4.5. has been entitled by the Authors as "Determination of essential oils. Composition of which essential oils have been specified by the Authors in section 4.5?”

Authors answer 7:

The title was corrected to ‘Sugar analysis’.

Comment 7:

“Please provide information about the solvent in which the essential oils were dissolved before they were applied to the paper discs in order to determine the zone of inhibition of the growth of the tested bacterial strains? (line 406)”

Authors answer 7:

    Lemongrass EO (1 µL) was added directly 6 mm paper discs. No solvent has been added.